# Incidence and predictive biomarkers of *Clostridioides difficile* infection in hospitalized patients receiving broad-spectrum antibiotics

Cornelis H. van Werkhoven [1✉], Annie Ducher[2], Matilda Berkell [3], Mohamed Mysara [3,4], Christine Lammens[3], Julian Torre-Cisneros [5], Jesús Rodríguez-Baño [6,7], Delia Herghea[8], Oliver A. Cornely [9,10,11], Lena M. Biehl[9,11], Louis Bernard[12], M. Angeles Dominguez-Luzon[13], Sofia Maraki [14], Olivier Barraud [15], Maria Nica[16], Nathalie Jazmati[17,18], Frederique Sablier-Gallis[2], Jean de Gunzburg [2], France Mentré[19], Surbhi Malhotra-Kumar [3], Marc J. M. Bonten[1,20], Maria J. G. T. Vehreschild [9,11,21✉] & the ANTICIPATE Study Group*

Trial enrichment using gut microbiota derived biomarkers by high-risk individuals can improve the feasibility of randomized controlled trials for prevention of *Clostridioides difficile* infection (CDI). Here, we report in a prospective observational cohort study the incidence of CDI and assess potential clinical characteristics and biomarkers to predict CDI in 1,007 patients ≥ 50 years receiving newly initiated antibiotic treatment with penicillins plus a beta-lactamase inhibitor, 3rd/4th generation cephalosporins, carbapenems, fluoroquinolones or clindamycin from 34 European hospitals. The estimated 90-day cumulative incidences of a first CDI episode is 1.9% (95% CI 1.1-3.0). Carbapenem treatment (Hazard Ratio (95% CI): 5.3 (1.7-16.6)), toxigenic *C. difficile* rectal carriage (10.3 (3.2-33.1)), high intestinal abundance of *Enterococcus* spp. relative to *Ruminococcus* spp. (5.4 (2.1-18.7)), and low Shannon alpha diversity index as determined by 16 S rRNA gene profiling (9.7 (3.2-29.7)), but not normalized urinary 3-indoxyl sulfate levels, predicts an increased CDI risk.

[1] Julius Center for Health Sciences and Primary Care, University Medical Center Utrecht, Utrecht University, Utrecht, the Netherlands. [2] Da Volterra, Paris, France. [3] Laboratory of Medical Microbiology, Vaccine & Infectious Disease Institute, University of Antwerp, Antwerp, Belgium. [4] Microbiology Unit, Environment Health and Safety, Belgian Nuclear Research Centre, SCK.CEN, Mol, Belgium. [5] Maimonides Institute for Research in Biomedicine of Cordoba (IMIBIC), Reina Sofia University Hospital, University of Cordoba (UCO), Cordoba, Spain. [6] Unidad Clínica de Enfermedades Infecciosas y Microbiología, Hospital Universitario Virgen Macarena, Sevilla, Spain. [7] Departamento de Medicina, Universidad de Sevilla, Instituto de Biomedicina de Sevilla (IBiS), Sevilla, Spain. [8] Oncology Institute Prof. Dr. I Chiricuta, Cluj Napoca, Romania. [9] Department I of Internal Medicine, Faculty of Medicine and University Hospital of Cologne, University of Cologne, Cologne, Germany. [10] University of Cologne, Faculty of Medicine and University Hospital Cologne, Chair Translational Research, Cologne Excellence Cluster on Cellular Stress Responses in Aging-Associated Diseases (CECAD), Cologne, Germany. [11] German Centre for Infection Research (DZIF), Partner Site Bonn-Cologne, Cologne, Germany. [12] Centre hospitalo-universitaire de Tours, Tours, France. [13] Hospital Universitari de Bellvitge, Universitat de Barcelona, Barcelona, Spain. [14] University Hospital of Heraklion, Heraklion, Greece. [15] Université Limoges, INSERM U1092, Centre Hospitalier Universitaire de Limoges, Limoges, France. [16] Infectious and Tropical Diseases Hospital "Dr. Victor Babes", Bucharest, Romania. [17] Institute for Medical Microbiology, Immunology and Hygiene, University of Cologne, Cologne, Germany. [18] Labor Dr. Wisplinghoff, Cologne, Germany. [19] Université de Paris, INSERM, IAME, Paris, France. [20] Department of Medical Microbiology, University Medical Center Utrecht, Utrecht University, Utrecht, the Netherlands. [21] Department of Internal Medicine, Infectious Diseases, University Hospital Frankfurt, Goethe University Frankfurt, Frankfurt am Main, Germany. *A list of authors and their affiliations appears at the end of the paper. ✉email: c.h.vanwerkhoven@umcutrecht.nl; maria.vehreschild@kgu.de

Clostridioides difficile infections (CDI) are considered the leading cause of infectious healthcare-related diarrhea worldwide and substantial costs have been attributed to its management[1–3]. Different medical interventions, including vaccines, probiotics, and agents neutralizing antibiotic residuals in the colon, are being developed to prevent CDI[4–7]. These interventions, however, require data from pivotal trials demonstrating clinical efficacy prior to approval by regulatory authorities. Given the relatively low incidence of CDI per individual patient in target populations, trials require large numbers of patients to demonstrate efficacy of a preventive measure. For example, a trial assessing the efficacy of probiotics for the prevention of CDI included 2981 patients, assuming a 12-week incidence of 4%. As the incidence of CDI in the placebo group was only 1.2%, the trial failed to demonstrate efficacy, and would have required a more than threefold larger sample size to reach adequate power for a 50% reduction of CDI[8]. The assumed incidence of the primary outcome is critical for successful execution of a trial and should be assessed in depth. Furthermore, the sample size can be reduced by enrichment of the RCT population with patients at a particularly high risk of CDI. Previous attempts to specify such a population identified advanced age, comorbidities, specific high-risk antibiotics (clindamycin, cephalosporins, fluoroquinolones, co-amoxiclav, carbapenems, and trimethoprim/sulfonamides), and previous colonization with C. difficile as potentially important predictive factors[9–11]. More recent findings suggest that the composition of the gut microbiota and their associated metabolites play an important role in the pathogenesis of CDI[12,13]. Therefore, further assessment of these potential biomarkers is warranted.

Here, we demonstrate that the 90-day incidence of CDI and antibiotic-associated diarrhea (AAD) in patients ≥ 50 years of age treated with predefined broad-spectrum antibiotic classes is 1.9% (95% CI: 1.1–3.0) and 14.1% (95% CI: 12.0–16.4), and that carbapenem treatment, toxigenic C. difficile carriage, and the composition and diversity of the gut microbiota predict CDI but not AAD.

## Results

A total of 1007 evaluable participants were enrolled (Fig. 1). Median age was 70 years (IQR: 62–79) and 592 participants (58.8%) were male (Table 1). Of 172 (17.1%) subjects who did not complete follow-up, 86 died after a median of 20 (IQR: 8–40) days and 86 withdrew their consent or were lost-to-follow-up after a median of 5 (IQR: 3–11) days.

Antibiotic treatment at day 1 and overall is provided in Table 2. The first dose of one of the five antibiotic classes was given after a median of 1 (IQR: 0–2) hospitalization day. Most participants (93%) received only one of the five antibiotic classes on day 1, most frequently a penicillin with a beta-lactamase inhibitor, followed by a third or fourth generation cephalosporin (Table 2). Apart from these five classes, 19% of patients received other additional antibiotics on day 1.

In total, 135 participants reported 176 AAD episodes, of which 114 episodes in 100 participants occurred within 28 days. The cumulative incidence of AAD within 28 and 90 days was 10.5% (95% CI: 8.6–12.5) and 14.6% (95% CI: 12.4–17.0), respectively. In 48 AAD episodes, no fecal sample was obtained for diagnosis of CDI, mainly because episodes were reported too late. Ten other subjects had fecal samples collected that were not correctly tested. The remaining 118 were tested according to the ESCMID guidelines, of which 5 were tested in the hospital laboratory only. Overall, 17 CDI episodes were detected in 15 subjects. Of these, nine first episodes occurred within 28 days. The number of subjects and CDI episodes per country and site are provided in

Supplementary Table 2. The median time to the first CDI episode was 18 days (IQR: 4–38; range: 1–78). The imputation model deviated from the planned analysis due to unforeseen imputation model overfit and indications for missing not at random (see Supplementary methods). After correction, the imputation model estimated that, among the non-tested AAD episodes, three CDI episodes were missed, of which two within 28 days. The estimated cumulative incidence of CDI, therefore, was 1.1% (95% CI: 0.6–2.1) within 28 days and 1.9% (95% CI: 1.1–3.0) within 90 days (Table 3).

Participants receiving carbapenems at day 1 had a higher incidence of CDI over the 90-days period than other patients (Table 3). Baseline toxigenic C. difficile carriage results were available in 983 (97.6%) participants, of which 35 were positive. The incidence of CDI was more than tenfold higher in participants with a positive baseline toxigenic C. difficile polymerase chain reaction (PCR) for the follow-up period of 28 and 90 days (Table 3). Time to first CDI was considerably shorter for participants with baseline toxigenic C. difficile carriage: 2 days (IQR: 1–3) vs. 30 days (IQR: 16–44) for those developing CDI but without baseline toxigenic C. difficile carriage (see Supplementary Table 4).

Baseline N3-IS levels were available for 938 (93.1%) participants. The distribution of N3-IS levels at baseline was comparable for patients developing or not developing AAD or CDI (Supplementary Table 3). None of the participants had a low baseline N3-IS level. No CDI episodes were identified in 28 participants with intermediate N3-IS levels. The AUC of baseline N3-IS for CDI within 90 days was 0.441 (95% CI: 0.323–0.559). No discriminative cut-off value for baseline N3-IS level for prediction of CDI was identified.

Day-6 N3-IS levels were available in 806 subjects of which 3 had CDI and 13 had AAD without known CDI status prior to the day-6 visit. As 6 subjects withdrew from the study at the day-6 visit, there were 784 subjects for this analysis. Day-6 N3-IS levels were lower in patients developing CDI compared to those not developing CDI (Supplementary Table 3). In 199 (25.4%) subjects with intermediate day-6 N3-IS levels, the cumulative incidence of CDI at day 90 was 4.1% (95% CI: 1.9–7.6), compared to 0.7% (95% CI: 0.2–2.0) in subjects with high day-6 N3-IS levels (SDHR: 5.8 (95% CI: 1.6–21.4)).

Baseline 16S rRNA gene profiling analysis results were available in 945 (93.8%) participants. A detailed description of the metagenomic analyses, including determination of alpha and beta diversity indices and LEfSe results, is reported in Berkell et al.[14]. For baseline alpha diversity, participants with a CDI episode had a median Shannon index of 2.19 (IQR: 1.87–3.37) and Inverse Simpson index of 5.53 (IQR: 3.12–17.33), while participants without a reported CDI episode had a median Shannon index of 3.38 (IQR: 2.91–3.76) and Inverse Simpson index of 13.87 (IQR: 8.74–21.26). Both indices were strongly predictive for CDI (Table 3) and moderately for AAD (Table 4).

The LEfSe analysis revealed two OTUs associated with CDI and 17 with no CDI (Supplementary Table 3 in Berkell et al.[14]), of which two and eight, respectively, had mean relative abundance of at least 0.5% and were selected for modeling. Based on model AICs and confirmation in the validation dataset, the best OTU-ratio model consisted of the relative abundance of Enterococcus (OTU1) divided by the relative abundance of Ruminococcus (OTU31), as in Table 3 (see Supplementary Table 5 for alternative models). Overall, 171 subjects (16.9%) were above the optimal breakpoint and had a predicted 90-day risk of CDI of 5.7%. For the OTU-abundance model, Enterococcus (OTU1) and Alistipes (OTU68) was selected as the best model (Table 3, Supplementary Table 6). 181 subjects (18.0%) had relative abundance of Enterococcus (OTU1) ≥ 0.087% and Alistipes (OTU68) < 0.087% and had a predicted 90-day CDI risk of 5.8%. The SDHR of the selected OTU models was similar to

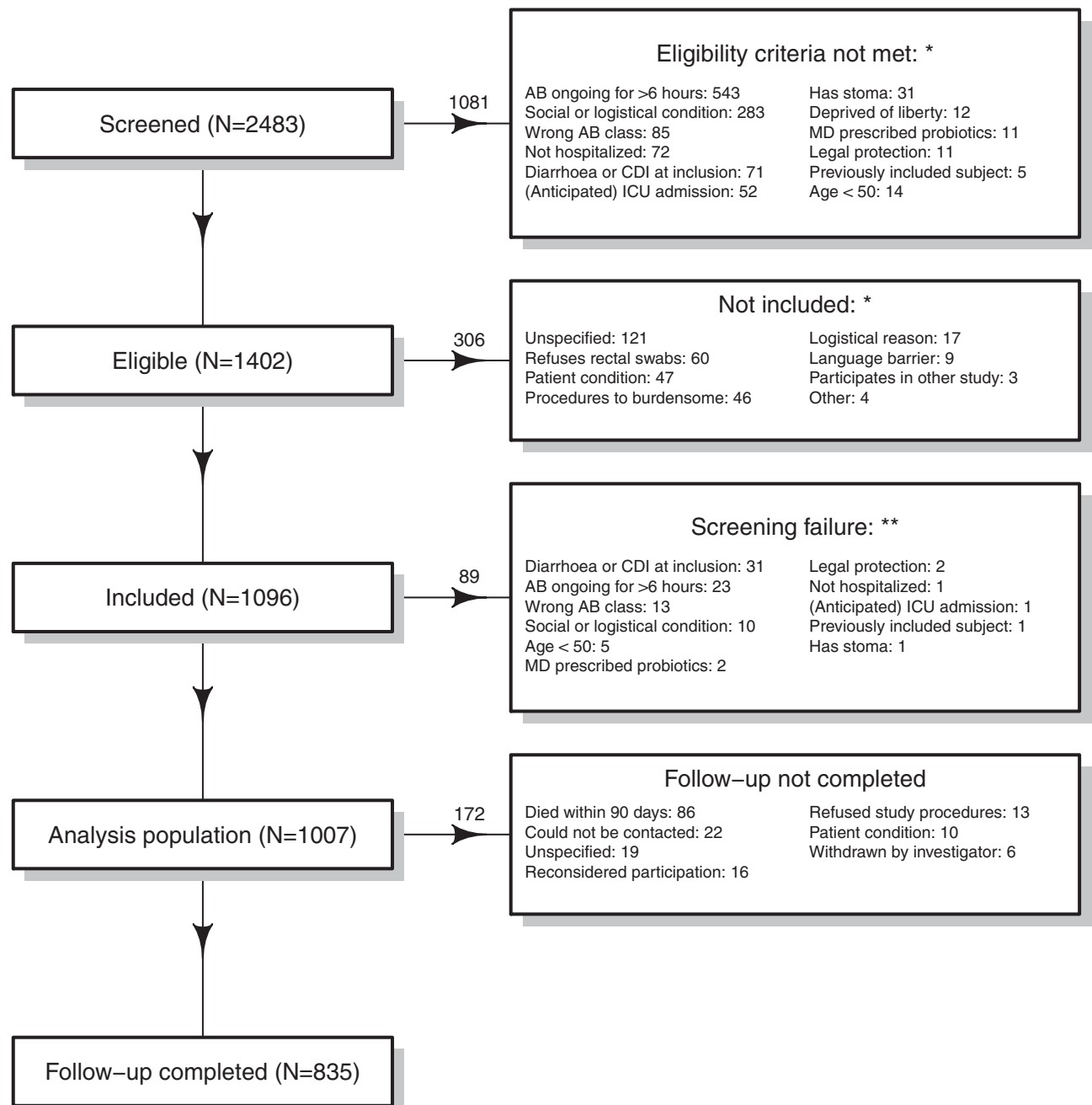

**Fig. 1 Inclusion flowchart.** *Based on 28 hospitals that provided screening data. Hospitals were requested to complete on the screening log for all screened patients up to March 23, 2017, and all eligible patients for the entire study period. Of the enrolled patients, 897 (89.1%) were enrolled in hospitals that provided screening data. **Subjects who signed informed consent but met one of the exclusion criteria at baseline. This includes 33 subjects from one side that applied an early consent procedure for subjects at high risk of receiving antibiotics in the near future. *Abbreviations*: AB antibiotics, CDI *C. difficile* infection, ICU intensive care unit, MD medical doctor.

the RR observed in the validation dataset: RR 4.6 (95% CI: 1.4–28.8) for the ratio of OTU1/OTU31 and RR 6.2 (95% CI: 2.0–34.8) for the relative abundance of OTU1 ≥ 0.087% and OTU68 < 0.087% (Supplementary Tables 5 and 6). Most but not all of the alternative OTU models were confirmed in the validation. For CDI within 28 days, predictive performance of 16S rRNA gene sequencing data could not be estimated due to the low number of CDI episodes within 28 days.

Different sensitivity analyses confirmed that rectal carriage of toxigenic *C. difficile* and the selected OTU models were predictive for CDI (Supplementary Tables 7–12). However, in the subset of countries with high testing rates, relative abundance of

*Porphyromonas* (OTU69) turned out to be more predictive as compared to OTUs selected in the main analysis. Carbapenem use at baseline was confirmed in some sensitivity analyses, but lost statistical significance in the subset of countries with high testing rates. Further exploration of the data revealed that CDI testing was correctly performed in 81% of AAD episodes in patients with baseline carbapenem use, compared to 65% of other AAD episodes. Due to the low number of CDI episodes observed, the original plan to perform multivariable analysis for CDI prediction was abandoned.

The incidence of AAD was increased threefold in subjects receiving carbapenems at baseline compared to patients not

**Table 1 Baseline characteristics.**

| Baseline variable | Total population (n = 1007) |
|---|---|
| Age (median [IQR]) | 70.0 [61.5, 79.0] |
| Male gender (%) | 592 (58.8) |
| Charlson comorbidity index (median [IQR]) | 5.1 [3.8, 6.8] |
| Karnofski score (median [IQR]) | 80 [70, 90] |
| Cardiovascular disease (%) | 321 (31.9) |
| Hematological/oncological disease (%) | 340 (33.8) |
| Diabetes mellitus (%) | 281 (27.9) |
| COPD (%) | 160 (15.9) |
| Gastrointestinal disease (%) | 156 (15.5) |
| Moderate to severe chronic kidney disease (%) | 132 (13.1) |
| Connective tissue disease (%) | 54 (5.4) |
| AIDS (%) | 11 (1.1) |
| Has history of CDI (%) | 14 (1.5) |
| Years since last CDI episode (median [IQR]) | 0.8 [0.3, 1.1] |
| Used systemic antibiotics in past 3 months (%) | 405 (42.0) |
| Used immunosuppressants in past 3 months (%) | 232 (23.2) |
| Used proton pump inhibitors in past 3 months (%) | 497 (50.9) |
| Hospitalized in past 12 months (%) | |
| Not hospitalized | 463 (46.8) |
| Once | 261 (26.4) |
| Twice | 102 (10.3) |
| Three or more times | 163 (16.5) |
| Received outpatient care in past month (%) | 556 (55.5) |
| Received tube feeding during current admission prior to inclusion (%) | 8 (0.8) |
| ICU admission in past 3 months (%) | 33 (3.3) |
| Indication for antibiotic treatment[a] | |
| Respiratory tract infection | 340 (33.8%) |
| Urinary tract infection | 171 (17.0%) |
| Abdominal infection | 157 (15.6%) |
| Prophylactic treatment | 90 (8.9%) |
| Febrile neutropenia | 85 (8.4%) |
| Skin and soft tissue infection | 66 (6.6%) |
| Other indications | 122 (12.1%) |

[a]Totals add up to >100% because some patients had more than one indication. Other indications include, bacteriuria (1), Bartonellosis (1), blood stream infection (15), cervical lymph node tuberculosis (1), central nervous system infection (7), endocarditis (13), infection with unknown focus (33), Lyme's disease (2), ocular infection (4), orchitis (1), osteoarticular infection (32), pericarditis (2), prosthesis infection (3), surgical site infection (5), Trauma (1), Vasculitis (2).
Abbreviations: AIDS acquired immunodeficiency syndrome, CDI Clostridioides difficile infection, COPD chronic obstructive pulmonary disease, ICU intensive care unit, IQR interquartile range.

receiving carbapenems at baseline (Table 4). Other antibiotic classes, history of CDI, toxigenic *C. difficile* carriage and N3-IS levels were not associated with an increased AAD incidence. The LEfSe analysis revealed 6 OTUs associated with CDI and 16 with no CDI (Supplementary Table 4 in Berkell et al.[14]), of which four and nine, respectively, had mean relative abundance of at least 0.5% and were selected for modeling. The ratio of the respective relative abundances of uncultured *Lachnospiraceae* (OTU21) over *Blautia* (OTU648) was selected as the best OTU ratio model (Table 4, see Supplementary Table 13 for AIC values). Subjects with this ratio ≥ 6 had a cumulative incidence of AAD within 90 days of 20.0% compared to 12.3% in the other subjects. The *C*-statistic of this model was 0.555, particularly due to a low sensitivity (Table 4 footnote). For the OTU-abundance model, low abundances of *Porphyromonas* (OTU69) and *Blautia* (OTU648) and high abundance of *Lachnospiraceae* (OTU21) best predicted an increased AAD risk, with similar predicted risks and *C*-statistic

as compared to the OTU ratio model (Table 4, Supplementary Table 13). The complete case analysis confirmed the association of carbapenems and OTU models with AAD (Supplementary Tables 14 and 15).

## Discussion

In this study of adults aged ≥ 50 years receiving broad-spectrum systemic antibiotic treatment during hospitalization, we observed a cumulative CDI incidence of 1.1% (95% CI: 0.6–2.1) within 28 days and 1.9% (95% CI: 1.1–3.0) within 90 days of antibiotic treatment initiation. Carbapenem prescription, rectal carriage of toxigenic *C. difficile* at baseline, a low alpha diversity of the intestinal microbiome, as well as a high relative abundance of *Enterococcus* (OTU1) and low relative abundance of *Ruminococcus* (OTU31) or *Alistipes* (OTU68) predicted an increased risk of CDI. Urinary N3-IS level at baseline was not associated with occurrence of CDI. AAD could not be predicted reliably with any of the studied parameters. The involvement of different infectious and non-infectious causes of AAD may hamper accurate biomarker-based prediction of AAD.

Although all evaluated antibiotic classes have been associated with CDI, in our study, only patients treated with carbapenems had a threefold increased risk of AAD and an almost fivefold increased risk of CDI within 90 days as compared to patients not receiving carbapenems at baseline. The higher incidence in patients receiving carbapenems may be due to the broader anti-microbial activity spectrum resulting in more disruption of the intestinal microbiota, or to differences in baseline risk factors in patients receiving carbapenems compared to those receiving other antibiotics. Independent of the mechanism, enrichment of a trial population with patients receiving carbapenems is likely to increase the incidence of CDI and hence make a trial more effi-cient. The expected efficiency gain will be driven by the pro-portion of patients receiving carbapenems, which was relatively low in our study, but varied substantially among hospitals. Few patients received clindamycin and CDI was observed in none, prohibiting incidence estimation for this antibiotic. Less than 2% of our population had a history of CDI, a risk factor for which we could not determine the incidence of CDI, due to the absence of outcomes in this small subgroup, even though CDI history has previously been found to predict a high CDI risk[15].

We studied three distinct biomarker types. Previous studies have found associations between urinary N3-IS levels and the microbiota composition. In particular, the relative abundance of bacteria from the class Clostridia was positively associated with N3-IS, whereas *Bacilli* were negatively associated with N3-IS[16,17]. As Clostridia have been associated with a decreased and *Bacilli* with an increased CDI risk, we hypothesized that patients with a low N3-IS level would be at an increased risk of CDI[18–21]. However, in our study baseline N3-IS level was not predictive. Yet, there was an association between low day-6 N3-IS level and an increased CDI incidence afterwards, confirming the direction of the hypothesized association. Altogether, we conclude that N3-IS is very unlikely to be a relevant baseline predictor of CDI risk in this patient population, but when measured after several days of antibiotic treatment it may predict the subsequent risk of CDI.

Toxigenic *C. difficile* carriage at baseline was detected in 3.6% of participants. In previous studies, the prevalence of toxigenic *C. difficile* carriage was very heterogeneous with a pooled pre-valence of 8.1% (95% CI: 5.7–11.1%)[11]. The low observed carriage prevalence may be due to a potentially lower sensitivity of rectal swabs compared to stool samples for detection of toxigenic *C. difficile* carriage. PCR testing based on swabs dipped in stool samples of CDI patients has yielded high detection rates, but this may be different for detection of colonization[22]. On the other

**Table 2 Antibiotic treatment at baseline and during follow-up.**

| Variable | N (%) D1 | N (%) any time | DOT |
|---|---|---|---|
| One antibiotic class[a] | 939 (93.2%) | 607 (60.3%) | |
| Two antibiotic classes[a] | 66 (6.6%) | 319 (31.7%) | |
| Three+ antibiotic classes[a] | 2 (0.2%) | 81 (8.0%) | |
| One antibiotic class[b] | 758 (75.3%) | 412 (40.9%) | |
| Two antibiotic classes[b] | 236 (23.4%) | 363 (36.0%) | |
| Three+ antibiotic classes[b] | 13 (1.3%) | 232 (23.0%) | |
| Penicillin with beta-lactamase inhibitor | 409 (40.6%) | 530 (52.6%) | 8 (5–12) |
| Third or fourth generation cephalosporin | 394 (39.1%) | 457 (45.4%) | 7 (4–9) |
| Fluoroquinolone | 181 (18.0%) | 326 (32.4%) | 9 (6–15) |
| Carbapenem | 64 (6.4%) | 143 (14.2%) | 8 (5–14) |
| Clindamycin | 29 (2.9%) | 45 (4.5%) | 8 (5–13) |
| Other antibiotic classes | 193 (19.2%) | 407 (40.4%) | 9 (5–17) |
| Any antibiotic | 1007 (100.0%) | 1007 (100.0%) | 11 (7–22) |

[a]Out of the five antibiotic classes given, not including other classes.
[b]Including other antibiotic classes.
Abbreviations: D1 day one of antibiotic treatment, DOT days on treatment given as Median (interquartile range); combination treatment is counted as one DOT.

**Table 3 Cumulative incidence of CDI within 28 and 90 days.**

| Subgroup | N | Events | Incidence | SDHR |
|---|---|---|---|---|
| *CDI within 28 days* | | | | |
| Total population | 1007 | 11.0 | 1.1% (0.6–2.1) | |
| Penicillin + BLI | 409 | 4.4 | 1.1% (0.4–2.7) | 1.0 (0.3–3.6) |
| Third/fourth gen cephalosporin | 394 | 3.5 | 0.9% (0.2–2.5) | 0.7 (0.2–2.8) |
| Fluoroquinolone | 181 | 1.6 | 0.9% (0.1–4.1) | 0.7 (0.1–5.1) |
| Carbapenem | 64 | 2.5 | 4.0% (0.8–11.8) | 4.2 (0.9–19.3) |
| Clindamycin | 29 | 0.3 | – | – |
| History of CDI | 31 | 0.5 | – | – |
| No history of CDI | 976 | 10.6 | 1.1% (0.6–2.1) | – |
| Toxigenic *C. difficile* carriage (GeneXpert, Cepheid) | 45 | 4.6 | 10.6% (3.2–23.2) | **16.2 (4.3–60.8)** |
| No toxigenic *C. difficile* carriage | 962 | 6.5 | 0.7% (0.3–1.6) | – |
| Intermediate N3-IS level (−8 ≤ log2(N3-IS) ≤ 2.5) | 38 | 0.2 | – | – |
| High N3-IS level (log2(N3-IS) > 2.5) | 969 | 10.8 | 1.2% (0.6–2.1) | – |
| Shannon index ≤ 2.586 | 164 | 7.2 | 4.6% (1.9–9.1) | **10.2 (2.3–44.4)** |
| Shannon index > 2.586 | 843 | 3.8 | 0.5% (0.1–1.4) | – |
| Inverse Simpson index ≤ 7.674 | 220 | 7.2 | 3.5% (1.5–6.9) | **7.4 (1.6–33.0)** |
| Inverse Simpson index > 7.674 | 787 | 3.8 | 0.5% (0.1–1.5) | – |
| *CDI within 90 days* | | | | |
| Total population | 1007 | 18.0 | 1.9% (1.1–3.0) | |
| Penicillin + BLI | 409 | 5.4 | 1.4% (0.5–3.1) | 0.6 (0.2–1.8) |
| Third/fourth gen cephalosporin | 394 | 7.1 | 1.9% (0.8–3.9) | 1.0 (0.4–2.8) |
| Fluoroquinolone | 181 | 2.7 | 1.6% (0.3–4.9) | 0.8 (0.2–3.3) |
| Carbapenem | 64 | 5.1 | 8.3% (2.7–17.7) | **5.7 (1.9–17.4)** |
| Clindamycin | 29 | 0.3 | – | – |
| History of CDI | 31 | 0.8 | – | – |
| No history of CDI | 976 | 17.1 | 1.9% (1.1–3.0) | – |
| Toxigenic *C. difficile* carriage (GeneXpert, Cepheid) | 45 | 4.9 | 11.4% (3.4–24.8) | **8.6 (2.7–27.5)** |
| No toxigenic *C. difficile* carriage | 962 | 13.1 | 1.5% (0.8–2.5) | – |
| Intermediate N3-IS level (−8 ≤ log2(N3-IS) ≤ 2.5) | 38 | 0.3 | – | – |
| High N3-IS level (log2(N3-IS) > 2.5) | 969 | 17.7 | 1.9% (1.1–3.1) | – |
| Shannon index ≤ 2.586 | 164 | 11.6 | 7.5% (3.7–13.0) | **9.7 (3.2–29.7)** |
| Shannon index > 2.586 | 843 | 6.4 | 0.8% (0.3–1.8) | – |
| Inverse Simpson index ≤ 7.674 | 220 | 11.6 | 5.7% (2.9–9.8) | **6.9 (2.3–21.3)** |
| Inverse Simpson index > 7.674 | 787 | 6.4 | 0.9% (0.3–2.0) | – |
| Ratio OTU1/OTU31 ≥ 8.5[a] | 171 | 11.5 | 5.7% (1.8–13.5) | **5.4 (2.1–18.7)** |
| Ratio OTU1/OTU31 < 8.5 | 836 | 6.4 | 1.1% (0.4–2.4) | – |
| OTU1 ≥ 0.087% and OTU68 < 0.087%[b] | 181 | 11.3 | 5.8% (2.3–11.8) | **5.4 (2.0–19.5)** |
| OTU1 < 0.087% OR OTU68 ≥ 0.087% | 826 | 6.7 | 1.1% (0.4–2.3) | – |

Based on the competing events model using multiple imputed data. The number of events is averaged over the imputation datasets and can therefore have decimals. Cumulative incidence was calculated using competing events analysis. Abbreviations: BLI beta-lactamase inhibitor, CDI Clostridioides difficile infection, CI confidence interval, N3-IS normalized 3-indoxyl sulfate level in urine, OTU1 Enterococcus, OTU31 Ruminococcus, OTU68 Alistipes, SDHR subdistribution hazard ratio. Bold text denotes statistical significance. [a,b] Bias-adjusted incidences and SDHRs are provided.
[a]The bias-adjusted sensitivity, specificity and C-statistic were 51.2%, 83.7%, and 0.675, respectively.
[b]The bias-adjusted sensitivity, specificity and C-statistic were 54.4%, 82.8% and 0.686, respectively.

**Table 4 Cumulative incidence of AAD within 28 and 90 days.**

| Subgroup | N | Events | Incidence | SDHR |
|---|---|---|---|---|
| *AAD within 28 days* | | | | |
| Total population | 1007 | 101.7 | 10.5% (8.7–12.6) | |
| Penicillin + BLI | 409 | 45.5 | 11.6% (8.6–15.0) | 1.2 (0.8–1.8) |
| Third/fourth gen cephalosporin | 394 | 29.6 | 7.8% (5.3–10.8) | **0.6 (0.4–1.0)** |
| Fluoroquinolone | 181 | 18.6 | 10.7% (6.6–15.9) | 1.0 (0.6–1.7) |
| Carbapenem | 64 | 15.6 | 25.0% (14.9–36.4) | **2.8 (1.7–4.9)** |
| Clindamycin | 29 | 2.0 | 7.6% (1.3–21.8) | 0.7 (0.2–2.6) |
| History of CDI | 31 | 3.8 | 13.0% (2.5–32.1) | 1.2 (0.3–4.6) |
| No history of CDI | 976 | 97.9 | 10.4% (8.5–12.5) | – |
| Toxigenic *C. difficile* carriage (GeneXpert, Cepheid) | 45 | 7.8 | 18.1% (7.9–31.4) | 1.9 (0.9–4.0) |
| No toxigenic *C. difficile* carriage | 962 | 93.9 | 10.2% (8.3–12.2) | – |
| Intermediate N3-IS level ($-8 \leq$ log2(N3-IS) $\leq 2.5$) | 38 | 2.8 | 7.5% (1.5–20.1) | 0.7 (0.2–2.7) |
| High N3-IS level (log2(N3-IS) > 2.5) | 969 | 98.9 | 10.6% (8.7–12.7) | – |
| Shannon index $\leq 3.155$ | 399 | 49.4 | 12.8% (9.6–16.5) | 1.5 (1.0–2.2) |
| Shannon index > 3.155 | 608 | 52.4 | 9.0% (6.8–11.5) | – |
| Inverse Simpson index $\leq 14.339$ | 529 | 59.2 | 11.6% (8.9–14.6) | 1.3 (0.8–1.9) |
| Inverse Simpson index > 14.339 | 478 | 42.5 | 9.3% (6.8–12.2) | – |
| *AAD within 90 days* | | | | |
| Total population | 1007 | 135.0 | 14.1% (12.0–16.4) | |
| Penicillin + BLI | 409 | 51.0 | 13.1% (9.9–16.6) | 0.9 (0.6–1.3) |
| Third/fourth gen cephalosporin | 394 | 47.0 | 12.6% (9.4–16.1) | 0.8 (0.6–1.1) |
| Fluoroquinolone | 181 | 25.0 | 14.6% (9.8–20.3) | 1.0 (0.7–1.6) |
| Carbapenem | 64 | 21.0 | 33.8% (22.3–45.7) | **3.0 (1.9–4.7)** |
| Clindamycin | 29 | 3.0 | 11.4% (2.8–26.8) | 0.8 (0.2–2.3) |
| History of CDI | 31 | 3.9 | 13.4% (2.6–33.0) | 0.9 (0.2–3.6) |
| No history of CDI | 976 | 131.1 | 14.1% (11.9–16.5) | – |
| Toxigenic *C. difficile* carriage (GeneXpert, Cepheid) | 45 | 8.3 | 19% (8.6–32.5) | 1.5 (0.7–3.1) |
| No toxigenic *C. difficile* carriage | 962 | 126.7 | 13.9% (11.7–16.2) | – |
| Intermediate N3-IS level ($-8 \leq$ log2(N3-IS) $\leq 2.5$) | 38 | 4.1 | 11.0% (3.0–24.7) | 0.8 (0.2–2.3) |
| High N3-IS level (log2(N3-IS) > 2.5) | 969 | 130.9 | 14.2% (12.0–16.6) | – |
| Shannon index $\leq 3.155$ | 399 | 67.2 | 17.7% (13.9–21.8) | **1.5 (1.1–2.2)** |
| Shannon index > 3.155 | 608 | 67.8 | 11.7% (9.2–14.5) | – |
| Inverse Simpson index $\leq 14.339$ | 529 | 80.1 | 15.9% (12.8–19.3) | 1.3 (0.9–1.9) |
| Inverse Simpson index > 14.339 | 478 | 54.9 | 12.1% (9.3–15.4) | – |
| Ratio OTU21/OTU648 $\geq 6^{\S}$ | 227 | 51.0 | 20.0% (14.7–25.8) | **1.7 (1.3–2.4)** |
| Ratio OTU21/OTU648 < 6 | 780 | 84.0 | 12.3% (9.9–15.2) | – |
| At least two of OTU69 < 0.140% or OTU21 $\geq$ 0.013% or OTU648 < 0.006%[a] | 448 | 87.2 | 18.4% (14.4–22.8) | **1.8 (1.4–2.4)** |
| Less than two of OTU69 < 0.140% or OTU21 $\geq$ 0.013% or OTU648 < 0.006% | 559 | 47.8 | 10.6% (8.2–13.4) | – |

Based on the competing events model using multiple imputed data. The number of events is averaged over the imputation datasets and can therefore have decimals. Cumulative incidence was calculated using competing events analysis. *Abbreviations: AAD* antibiotic associated diarrhea, *BLI* beta-lactamase inhibitor, *CDI Clostridioides difficile* infection, *CI* confidence interval, *N3-IS* normalized 3-indoxyl sulfate level in urine, *OTU21* uncultured *Lachnospiraceae*, *OTU69 Porphyromonas*, *OTU648 Blautia*, *SDHR* subdistribution hazard ratio. Bold text denotes statistical significance. [§,†]Bias-adjusted incidences and SDHR are provided. [§]The bias-adjusted sensitivity, specificity and *C*-statistic were 32.1%, 79.0%, and 0.555, respectively. [a]The bias-adjusted sensitivity, specificity and *C*-statistic were 58.2%, 57.8% and 0.580, respectively.

hand, the relative risk of toxigenic *C. difficile* carriage for CDI is high compared to previous studies in which relative risks between 2.5 and 10 have been observed[11]. The very short time to CDI occurrence in participants colonized with *C. difficile* was surprising. A recent study among hospitalized medical patients also using PCR testing on rectal swabs for detection of toxigenic *C. difficile* found a similar carriage rate of 3.4% and a relative risk for CDI among carriers of 16.6 (95% CI: 4.0–69.1), but no difference in time to occurrence of CDI (median [range] 23 [4–30] days for carriers vs. 11 [4–30] days for noncarriers)[23]. The low prevalence of *C. difficile* carriage would make this biomarker very inefficient for enrichment of prevention trials, as about 30 patients would need to be screened to enroll one high risk patient. Moreover, rapid point-of-care testing must be available with affordable cost.

Concerning the 16S rRNA analysis, *Enterococcus* (OTU1), *Ruminococcus* (OTU31), or *Alistipes* (OTU68) were most predictive of the risk of CDI. Participants with more or equal than 8.5 times higher relative abundance of *Enterococcus* compared to

*Ruminococcus* had a fivefold higher risk to develop CDI. Patients with high relative abundance of *Enterococcus* and low relative abundance of *Alistipes* also had an increased risk of approximately 5 times. Several alternative models with similar performance were identified. There is large uncertainty around these estimates, with high risk of false positive findings, due to the low number of CDI episodes and substantial expected bias that needed to be corrected for. Therefore, we validated our findings in an independent dataset and selected from the best performing models those that were confirmed in the validation dataset.

High abundance of *Enterococcaceae* spp. has previously been associated with CDI risk[18]. *Ruminococcus* spp. are a member of the class Clostridia, which have repeatedly been associated with a decreased risk of CDI[18–21]. *Alistipes* spp. are a member of the phylum Bacteroidetes which have also been associated with CDI in two of these studies[18,19]. We fitted two different types of models, which had comparable performance, but may have different applications in practice. The identified ratio of

*Enterococcus/Ruminococcus* could be translated into a duplex quantitative PCR that could serve as a rapid point-of-care screening test to identify patients at high risk of CDI and to enrich trial populations. The design of such a test would require further testing and optimization and independent confirmation of the predictive performance. Unfortunately this could not be performed in the current study due to lack of leftover material. The OTU-coefficients model may be more useful if relative abundances are generated, such as with 16S rRNA gene analysis or similar techniques. Recent advances in high-throughput sequencing, such as nanopore technology may allow the point-of-care application of this model or rapid estimation of alpha diversity in the future. At present, however, the costs associated with such analyses would probably not outweigh the benefits of more accurate risk prediction.

Several limitations of the study have to be discussed. First, we lacked a CDI test result in one third of AAD episodes, mostly due to delayed reporting of diarrhea by the participant, especially after hospital discharge. The apparent difference in testing between antibiotic groups may have influenced the association with carbapenem use and lack of association with other antibiotic classes. If CDI causes more severe diarrhea symptoms compared to non-CDI AAD, patients with CDI may have reported diarrhea more frequently, thus fewer CDI episodes might be missing. To our knowledge, however, there is no published data to support this and we did not collect data on diarrhea severity. Apparently, the collection of information through a daily paper diary was insufficient to keep participants aware of the requirement to report diarrheal episodes. For future studies, more active follow-up procedures should be considered, such as a smartphone app, repeated text messages, or regular telephone calls. We mitigated the problem of missing samples by performing multiple imputation and we performed different sensitivity analyses to test the robustness of the results, which yielded similar results. Second, the low number of CDI episodes in our study precluded multivariable analysis. Hence, whether carbapenem use, *C. difficile* carriage, and the selected OTUs are independent predictors remains to be determined. Third, 9% of included participants were lost for other reasons than death, of which most withdrawals occurred around day 6. The collection of another rectal swab and urine sample during the second visit apparently decreased the willingness of participants to continue study participation. We solved this analytically by applying a survival model, which allows censoring of follow-up times. However, this model assumes uninformative censoring, which cannot be substantiated with data.

Due to the low incidence of CDI, a trial aiming to prevent CDI in this target population will require a substantial number of subjects. For example, with a baseline 28-day CDI incidence of 1.2% and a relative reduction of 50%, 5205 patients per arm are needed to achieve 90% power at a two-sided significance level of 0.05. Using a risk factor (e.g., a biomarker) to enrich the trial with high-risk patients reduces the number of patients for randomization and follow-up. At the same time, the number of patients that must be screened for the risk factor will most likely increase. The ideal risk factor has a fairly high prevalence in the population (e.g., 20 or 30%) and is strongly associated with the outcome of interest. The OTU-based risk prediction of CDI meets these criteria. Yet, the feasibility of such risk prediction also depends on the ease and cost of screening. A biomarker is more expensive to measure than comorbidities, hence even if the biomarker would constitute a stronger risk factor, it may be less efficient to use. Potentially, a multivariable approach using patient history, demographics, antibiotic exposure and clinically available laboratory test results would be as good as the biomarkers tested in our study. Alternatively, the biomarkers could be of added

value in patients with an intermediate predicted risk based on clinically available predictors. All of this could not be tested in this study due to the low number of CDI episodes.

In conclusion, the 28 and 90-day incidences of CDI in patients ≥50 years receiving broad-spectrum antibiotics during hospitalization were 1.14% and 1.89%, respectively. Carbapenem treatment, *C. difficile* carriage, and high relative abundance of *Enterococcus* spp. vs. low relative abundance of *Ruminococcus* spp. or *Alistipes* spp. at baseline predicted occurrence of CDI, the latter being confirmed in an independent validation dataset with similar relative risks despite being from a different design and setting.

## Methods

**Study design**. The "AssessmeNT of the Incidence of *C. difficile* Infections in hospitalized Patients on Antibiotic TrEatment" (ANTICIPATE) study was an international multicenter prospective observational cohort study in 34 hospitals (21 university and 13 nonuniversity hospitals) from France, Germany, Greece, the Netherlands, Romania and Spain. Patients were recruited from September 2016 through October 2017. The study protocol was approved by a central ethics review board in each country and/or the local institutional review boards of each hospital, in accordance with the local regulations (Supplementary methods: Ethics committees). The study was registered at ClinicalTrials.gov: NCT02896244.

**Eligibility and recruitment**. Pharmacy registries or admission lists of hospitalized patients were screened for eligibility during office hours. Patients were eligible if they met the following inclusion criteria: (1) hospitalization at the time of enrollment, (2) age ≥ 50 years, (3) initiation of treatment less than 6 h before enrollment or scheduled treatment within the next 72 h of intravenous or oral treatment with an intended duration ≥5 days (≥1 day for clindamycin) with at least one of the following antibiotic classes: third or fourth generation cephalosporins, fluoroquinolones, penicillins with beta-lactamase inhibitors, clindamycin, or carbapenems. Exclusion criteria were: (1) ongoing antibiotic treatment with one of the above classes initiated >6 h before inclusion into the study, (2) ICU admission at the time of inclusion or anticipated ICU admission within 48 h, (3) suspected or diagnosed CDI, ongoing treatment for CDI, or diarrhea at the time of inclusion, (4) stoma, (5) previous inclusion into the study, (6) ongoing probiotic treatment for prevention of CDI, (7) social or logistical condition which in the opinion of the investigator might interfere with the conduct of the study, (8) subject to legal protection, and (9) deprived of liberty by judicial or administrative decision. Written informed consent was obtained from all participants prior to any study related procedures.

**Outcome definitions**. The primary endpoint was CDI within 28 days after initiation of antibiotic treatment. An episode of CDI was defined as a clinical picture compatible with CDI and microbiological evidence of free toxins and the presence of *C. difficile* in stools, using one of the two algorithms recommended in the 2016 European Society of Clinical Microbiology and Infectious Diseases (ESCMID) guideline, without reasonable evidence of another cause of diarrhea, or pseudomembranous colitis as diagnosed during endoscopy, after colectomy or on autopsy[24]. Diarrhea was defined as loose stools, i.e., taking the shape of the receptacle, corresponding to Bristol stool chart types 5–7, and a stool frequency of at least three in 24 consecutive hours. Secondary endpoints included incidence of CDI within 90 days and incidence of AAD within 28 and 90 days.

**Data and samples**. During the enrollment visit, a urine sample for normalized 3-indoxyl sulfate (N3-IS, expressed as nmol 3-IS/μmol creatinine), a rectal swab for 16S rRNA gene profiling, and a rectal swab for *C. difficile* carriage detection by PCR (Cepheid GeneXpert) were collected. At study day 6 ± 1 (or day 3 or 4 in case of early discharge), another urine sample for N3-IS measurement and rectal swab for 16S rRNA profiling were collected. Data on baseline demographics and comorbidities, use of antibiotics and laxatives, as well as diarrhea episodes during hospitalization where retrieved from medical records. In addition, participants were asked to complete a diary on a daily basis assessing presence of diarrhea, use of antibiotics, and of laxatives throughout the 90-day follow-up period. At the end-of-study visit after study day 90 (either a phone call or a bed-side visit), the diary was collected and a final inquiry made about potentially missed diarrheal episodes. If participants reported diarrhea during the 90-day follow-up period, an additional fresh fecal sample was collected and tested for CDI in a national central laboratory. The hospitals' routine fecal collectors were used for episodes occurring during hospitalization and participants received a stool collection system for episodes occurring after discharge. See supplementary methods for detailed description of sample collection and processing.

**Sample size calculation**. The primary aim of this study was to estimate the incidence of CDI, which would be the main determinant of sample size calculations

of any RCT targeting the prevention of CDI in this at-risk population. The sample size was chosen such that the impact of randomly over- or underestimating the incidence would be acceptably low. Assuming an event rate of approximately 2% in the total included population, 839 participants would be needed to accurately determine the event rate. Assuming an event rate of 5% in one of the subgroups, 335 participants would be needed in that subgroup to accurately determine the event rate in the high-risk group. If one third of participants were part of such a high-risk population, this would require a total study population of 1005 patients. Therefore, we aimed to include 1000 participants into the study. This sample size was deemed sufficient to perform a robust sample size calculation for an RCT.

**Data analysis.** Data were analyzed in R version 3.5.1 (2018-07-02)[25]. Baseline characteristics, antibiotic use, and sample collection data are given for the total population as proportions, mean (SD), or median (IQR), as appropriate.

Missing CDI test results from the central laboratory were complemented with the local laboratory results, if tested according to one of the ESCMID algorithms. For other missing data, we used the *mice* package for multiple imputation with 100 imputed datasets (see Supplementary methods). We used Rubin's rule to derive confidence intervals.

16S rRNA gene sequencing was performed as described in Berkell et al.[14]. Potential predictive biomarkers that emerged from the linear discriminant analysis effect size (LEfSe) model performed for CDI vs. no CDI and AAD (including CDI) vs. no AAD at the level of the operational taxonomic unit (OTU) were utilized for further analysis here. OTUs were defined by a cut-off of 97% similarity of the reads which is supposed to resemble species boundaries, although imperfect[26].

To calculate the incidences of CDI and AAD at day 28 and day 90, cumulative incidences were estimated from the imputed data using the cumulative incidence function with death as a competing event (*cmprsk* package version 2.2–7)[27]. This method takes loss to follow-up and competing events (death) into account. Stratified cumulative incidences were estimated per subgroup and were compared using subdistribution hazard ratios (SDHR) from the competing events model. The SDHR is calculated as the ratio of the subdistribution hazard function, which is defined as the instantaneous rate or events of a certain type in subjects who have not yet experienced that event[28] and can be interpreted as a relative risk if the incidence is low. Predefined subgroups were antibiotic class at baseline, history of CDI, baseline toxigenic *C. difficile* carriage, and low, intermediate and high N3-IS levels at baseline.

Stratified incidences for baseline N3-IS levels were first analyzed using previously published breakpoints and categorized as low (log2(N3-IS) < −8), intermediate (−8 ≤ log2(N3-IS) ≤ 2.5), or high (log2(N3-IS) > 2.5)[16]. Next, we calculated the area under the curve (AUC) of the receiver operating characteristic (ROC; *pROC* package version 1.13.0) and determined the optimal breakpoint based on the Youden index (29; *OptimalCutpoints* package version 1.1–3), assuming a lower value would be associated with a higher CDI risk. We also performed an unplanned analysis of CDI incidences at day 90 stratified for day-6 N3-IS levels (using the previously published breakpoints) in patients with no CDI prior to the day-6 visit and with day-6 N3-IS results available.

The microbiome-based risk strata were not prespecified. We defined low and high alpha diversity based on the Shannon index and Inverse Simpson index, for which we used the Youden index for dichotomization[29]. From the OTUs associated with the studied outcome identified by LEfSe[14], we further selected those with mean relative abundance of at least 0.5%. Based on these OTUs we defined two kinds of biomarkers. The first group of biomarkers were the ratio of the respective relative abundances of one positively and one negatively associated OTU (termed OTU ratio). For the negatively associated OTU, counts of zero were set to 0.5 to avoid division by zero. We used the Youden index for dichotomization[29]. The dichotomized OTU ratios were tested in the competing events model and ranked by Akaikes Information Criterion (AIC). The second group of biomarkers were OTU relative abundances, dichotomized based on the Youden index. These were tested in the competing event model as separate coefficients in a step-wise forward selection procedure based on the AIC using a selection threshold of −2 points. For both model selection approaches, we performed bootstrap cross-validation to determine the bias-corrected incidences, SDHR, sensitivity, specificity and *C*-statistic. The OTU model results for CDI were subsequently externally validated using data from a previously published nested case–control study[18]. In short, of 599 hospitalized adult patients prospectively enrolled in this Canadian study and followed until 60 days after discharge, 31 developed CDI, of which fecal samples collected prior to onset of CDI were available in 25 patients. These were matched to 25 non-CDI patients on sex, age (±5 years) and date of hospitalization (±2 months). Raw reads were retrieved from the corresponding author and processed to align with our analysis as described by Berkell et al.[14]. One CDI case with only 32 remaining reads was excluded from the analysis. For calculation of incidences and relative risks (RR), we applied weights in accordance to the sampling fraction of cases and controls and performed bootstrapping to derive 95% confidence intervals. Among the candidate OTU models derived from the ANTICIPATE data and that also predicted CDI in the validation dataset, the models with the lowest AIC value were selected as best predictive model.

The following sensitivity analyses were performed for CDI: (1) using non-imputed data assuming that missing CDI test results were negative (i.e., cumulative incidence of observed CDI); (2) complete case analysis, excluding participants with AAD but missing CDI test results; (3) restricted to countries with at least 75% of AAD episodes tested for CDI.

**Reporting summary.** Further information on research design is available in the Nature Research Reporting Summary linked to this article.

## Data availability

16S rRNA and shotgun metagenomic sequence data generated and analyzed in this study have been deposited in the NCBI Sequence Read Archive with the accession code PRJNA685914. Human reads were identified and removed prior to shotgun metagenomics data upload. All other data generated in this study are available from the corresponding author upon reasonable requests; access to clinical data is restricted as the informed consent provided does not allow for open publication of these data. Raw data from Vincent et al.[18] utilized as a validation cohort in this study was kindly provided by Prof. Amee Mangees (University of Brittish Columbia, Vancouver, Canada).

## Code availability

An example dataset and analysis code is provided as supplementary information. Full analysis code can be requested from the corresponding author.

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

## Acknowledgements

This research project was supported by the Innovative Medicines Initiative Joint Undertaking (IMI JU) under grant agreement No. 115523, the Combatting Bacterial Resistance in Europe (COMBACTE) consortium, resources of which are composed of financial contribution from the European Union Seventh Framework Program (FP7/2007–2013) and Da Volterra, as a member of the European Federation of Pharmaceutical Industries and Associations (EFPIA) companies in kind and cash contribution. We thank Prof. Amee Manges (University of British Columbia, Vancouver, Canada) for providing the validation dataset utilized in this study.

## Author contributions

C.Hv.W. contributed to the design, coordination, and data analysis of the study and wrote the first draft of the paper. A.D., Jd.G., and M. Bon contributed to the design of the study and interpretation of the data. M. Ber, M.M., and S.M.K. contributed to the analysis and interpretation of 16S rRNA gene profiling. C.L. contributed to the coordination of the study. J.T.C., J.R.B., D.H., O.A.C., L.M.B., L.B., M.A.D., S.M., O.B., M.N., and N.J. contributed to the collection of data and samples. F.S.G. and F.M. contributed to the interpretation of the data. M.V. contributed to the design and coordination of the study and supervised the writing of the paper. All authors critically reviewed the paper.

## Competing interests

C.Hv.W. received speaker fees from Pfizer and Merck/MSD, and non-financial research support from bioMérieux. A.D. and F.S. are employees and shareholders of Da Volterra, Paris. J.G. is a consultant and shareholder of Da Volterra. J.T.C. has received speaker and consultant fess from Pfizer, MSD, Shionogy, Menarini and research support from Pfizer. O.A.C. is supported by the German Federal Ministry of Research and Education, is funded by the Deutsche Forschungsgemeinschaft (DFG, German Research Foundation) under Germany's Excellence Strategy—CECAD, EXC 2030—390661388 and has received research grants from Actelion, Amplyx, Astellas, Basilea, Cidara, Da Volterra, F2G, Gilead, Janssen, Medicines Company, Melinta, Merck/MSD, Octapharma, Pfizer, Scynexis, is a consultant to Actelion, Allecra, Amplyx, Astellas, Basilea, Biosys, Cidara, Da Volterra, Entasis, F2G, Gilead, Matinas, MedPace, Menarini, Merck/MSD, Mylan, Nabriva, Noxxon, Octapharma, Paratek, Pfizer, PSI, Roche Diagnostics, Scynexis, and Shionogi, and received lecture honoraria from Al-Jazeera Pharmaceuticals, Astellas, Basilea, Gilead, Grupo Biotoscana, Merck/MSD and Pfizer. L.M.B. has received lecture honoraria from Astellas and Merck/MSD, and travel grants from 3M and Gilead. OB received speaker fees and/or travel grants from Pfizer, MSD, Roche and Sanofi and has been a consultant to bioMérieux and Mylan. F.M. is a consultant for Da Volterra, IPSEN, Servier and received research grants from Da Volterra, Sanofi and Servier. MJGTV has received research grants from 3M, Astellas Pharma, Da Volterra, Gilead Sciences, Glycom, MaaT Pharma, Merck/MSD, Organobalance, Seres Therapeutics; speaker fees from Astellas Pharma, Basilea, Gilead Sciences, Merck/MSD, Organobalance, Pfizer and has been a consultant to Alb Fils Kliniken GmbH, Astellas Pharma, Bio-Mérieux, Da Volterra, Ferring, MaaT Pharma, Merck/MSD. The remaining authors declare no competing interests.

## Additional information

## the ANTICIPATE Study Group

Annemarie M. S. Engbers[1], Marieke J. A. de Regt[22], Herman Goossens[3], Basil Britto Xavier[3], Marie-Noelle Bouverne[2], Pieter Monsieurs[4], Uta Merle[23], Andreas Stallmach[24], Jan Rupp[25], Johannes Bogner[26], Christoph Lübbert[27], Gerda Silling[28], Oliver Witzke[29], Achilleas Gikas[14], George Daikos[30], Sotirios Tsiodras[31], Athanasios Skoutelis[32], Helen Sambatakou[33], Miquel Pujol[13], Jose M. Aguado[34], Emilio Bouza[35], Javier Cobo[36], Benito Almirante[37], Simin A. Florescu[16], Andrei Vata[38], Adriana Hristea[39], Mihaela Lupse[40], Deborah Postil[15],

Jean-Michel Molina[41], Victoire De Lastours[42], Thomas Guimard[43], Jean-Philippe Talarmin[44], Xavier Duval[45] & Odile Launay[46]

[22]Department of Internal Medicine and Infectious Diseases, University Medical Center Utrecht, Utrecht, the Netherlands. [23]Universitätsklinikum Heidelberg, Heidelberg, Germany. [24]Universitätsklinikum Jena, Jena, Germany. [25]Universitätsklinikum Schleswig-Holstein, Lübeck, Germany. [26]Klinikum der Universität München, München, Germany. [27]Universitätsklinikum Leipzig, Leipzig, Germany. [28]Universitätsklinikum Aachen, Aachen, Germany. [29]Universitätsklinikum Essen, Essen, Germany. [30]Laiko General Hospital, Athens, Greece. [31]University General Hospital ATTIKON, Athens, Greece. [32]Evangelismos General Hospital, Athens, Greece. [33]Ippokratio Hospital, Athens, Greece. [34]Hospital Universitario 12 de Octubre, Madrid, Spain. [35]Hospital Universitario Gregorio Marañón, Madrid, Spain. [36]Hospital Universitario Ramón y Cajal, Madrid, Spain. [37]Hospital Universitari Vall d'Hebrón, Barcelona, Spain. [38]Clinical Hospital of Infectious Diseases of Iasi, Iasi, Romania. [39]The National Institute of Infectious Diseases Matei Bals, Bucharest, Romania. [40]Cluj Napoca Infectious Disease Clinical Hospital, Cluj Napoca, Romania. [41]Hôpital St Louis, Paris, France. [42]APHP Beaujon, Paris, France. [43]Centre Hospitalier Départemental Vendée, La Roche sur Yon, France. [44]CH de Cornouaille, Quimper, France. [45]APHP Bichat, Paris, France. [46]APHP Hôpital Cochin, Paris, France.

