## [Peer Review File · Nature Communications]

REVIEWER COMMENTS

Reviewer #1 (Remarks to the Author):

This manuscript is a part of a head to head publication. I find this problematic as both publications basically use the same results to arrive at the same conclusion for the microbiology part, that *Enterococcus* is longitudinally associated with CDI. As I see it the papers should have been merged to one paper. Basically, that would include expansion of the microbiological part in the current manuscript. Alternatively, the microbiological part should be removed. Since this is a multicenter study I think the differences across nations in CDI positive samples should be addressed in more depth.

I agree that the use of prospective multicenter studies in identifying markers predictive of *Clostridium difficile* infections is very valuable. However, in order to be predictive the predictive power have to be independently validated. Furthermore, the study was initially powered to determine the CDI incidence, not for identifying predictive biomarkers.

Information I feel is lacking:

- Distribution of CDI incidences across study sites
- Underlying diseases for the antibiotic treatment
- Nation-wide differences

Detailed comments:

I did not find a reference to Suppl tabl1 S1. I think this a really important table.

I. 69-74 I don't thing the underlying measures for the hazard ratios are easily accessible to the reader

Table 3 and 4. Although this is not my specialty field, but were multiple testing's considered in determining the statistical significance? How were thresholds determined, and how would this affect the statistical tests? Did you use cross-validation?

I. 177-182 I really find this problematic as the same results are published twice. I would have liked to see the results in one place.

L 194 Do not use the OTU subscript notion. This is not common.

I. 278-280 These findings need to be validated as they are based on a limited number of CDI patients. I think this can be done for the same hospitals as included.

I. 286-288. I doubt this would be possible, but if so then this test should have been used to validate the findings in the current work.

I. 327-328 I agree with the authors the low CDI incidence rate makes it difficult with prospective studies, including the risk of overfitting data based on few patients. I definitely think the study is big and valuable, but I am skeptical that the identified biomarkers are truly predictive in a general sense.

Reviewer #2 (Remarks to the Author):

This is a large, multi-center prospective cohort study (namely the ANTICIPATE study) that investigates the microbial signatures in patients with *Clostridioides difficile* infection (CDI). The overall goal of this project is to identify unique baseline biomarkers that predict the risk of CDI before receiving antibiotics.

The authors recruited 1,007 patients who underwent antibiotic therapy during hospitalization. Patients were followed by for 90 days after infusion. Microbial (rectal swab and stool) samples were collected before/after antibiotics treatment and at the onset of a diarrhea episode. Of those, 1.14% and 1.89% of patients subsequently developed CDI within 28 days or 90 days, respectively. The patients who subsequently develop CDI had lower microbial diversity at the baseline. Perhaps not surprisingly, CDI patients carry toxigenic *C. difficile* before the onset of CDI. Among antibiotics, carbapenem prescription correlated with the incidence of CDI. Moreover, the ratio of *Enterococcus/Blautia* in the baseline microbiota was increased in patients who subsequently develop CDI. These parameters were not associated with the risk of non-CDI antibiotic-associated diarrhea (AAD), indicating that those are selective biomarkers for the prediction of CDI.

General comment:

Overall this is a well-performed clinical study that highlights potential biomarkers that can help predict the risk for CDI. In particular, microbial signatures associated with the risk of CDI are interesting. However, this study is correlative/descriptive, and hence the impact on the broader readership, particularly the basic science community, may be limited. Therefore, this manuscript is more suitable for a subspecialty clinical journal.

Specific comments:

The authors nicely showed microbial biomarkers that may predict the risk of CDI before antibiotics treatment. The carriage of toxigenic *C. difficile* is not surprising, as this bacterium causes CDI. The increase in the ratio of *Enterococcus*/*Blautia* is interesting. However, the specificity and sensitivity of this microbial signature are not validated. It would be nice if the authors could validate their findings with other cohorts.

REVIEWER COMMENTS

Reviewer #1:

This manuscript is a part of a head to head publication. I find this problematic as both publications basically use the same results to arrive at the same conclusion for the microbiology part, that *Enterococcus* is longitudinally associated with CDI. As I see it the papers should have been merged to one paper. Basically, that would include expansion of the microbiological part in the current manuscript. Alternatively, the microbiological part should be removed.

Response: we have split the study results in two manuscripts for two reasons. First, the purpose of the two manuscripts is different, whereas manuscript NCOMMS-20-22939 has the objective to elucidate the biological mechanisms that explain the occurrence of CDI, the current manuscript translates these findings into a risk prediction model applicable for individual patients. The current manuscript also applies a similar approach for other risk factors. Combining the two manuscripts would require substantial expansion of the methods, results and discussion section of the paper. Second, we consider the readership of the two articles to be distinct, being fundamental researchers for the first and translational/clinical researchers for the second paper. We have avoided as much as possible repetition of the methods and results and refer to each other to clarify to the readers that the results are obtained from the same study.

Since this is a multicenter study I think the differences across nations in CDI positive samples should be addressed in more depth.

Response: We have added an overview of the number of enrolments and CDI episodes per site / country in the supplementary appendix. Unfortunately the low number of CDI episodes precludes a reliable stratification by country.

I agree that the use of prospective multicenter studies in identifying markers predictive of *Clostridium difficile* infections is very valuable. However, in order to be predictive the predictive power have to be independently validated. Furthermore, the study was initially powered to determine the CDI incidence, not for identifying predictive biomarkers.

Response: We agree thank the reviewer for this suggestion. We have now performed independent validation of the results using data from a previously published nested case-control study. Please see methods and results section for details. In short, the study includes 25 CDI patients and 25 randomly selected controls from a prospective cohort study. The methods for 16S rRNA gene profiling of fecal samples collected at baseline were compatible with our study. The validation confirmed most of the candidate OTU models identified in our study. Some models were not confirmed, although the wide confidence interval of the relative risk estimates does not exclude an association either. We now selected the model based on two criteria, 1) being statistically significant in both datasets and 2) among these, the model that had the lowest AIC value. The OTU ratio model as well as the OTU abundance model performed similarly in the validation dataset as compared to our study.

The study was powered to determine the CDI incidence *in a given subgroup* and can therefore also identify predictive biomarkers. The fact that several predictive biomarkers were statistically significant and now also confirmed independently, underlines that the study had sufficient power for identification of these biomarkers.

Information I feel is lacking:

- Distribution of CDI incidences across study sites
- Nation-wide differences

Response: Although we agree this would yield interesting data, the low number of CDI episodes unfortunately precludes a reliable stratification of the results by country. We have added the number of inclusions and number of CDI episodes per site and per country to the supplementary appendix. Since the enrolled population differed by site, for logistical reasons such as which wards had capacity to enrol patients, these data should be interpreted with caution.

- Underlying diseases for the antibiotic treatment

Response: The indications for antibiotic treatment are now added to the baseline table.

Detailed comments:

I did not find a reference to Suppl tab1 S1. I think this a really important table.

Response: these data are presented to support the multiple imputation methods description, which is referenced to as 'supplementary methods' (also including Supplementary Figures S1-S3). We have now provided a new table with number of inclusions and CDI endpoints per country and per site (Supplementary Table S2) and refer to this in the results section.

I. 69-74 I don't think the underlying measures for the hazard ratios are easily accessible to the reader

Response: The size of the abstract does not allow an explanation of the hazard ratio, we have now added this in the methods section.

Table 3 and 4. Although this is not my specialty field, but were multiple testing's considered in determining the statistical significance? How were thresholds determined, and how would this affect the statistical tests? Did you use cross-validation?

Response: We have not corrected for multiple testing. Cross-validation was used as described in line 433-435 of the methods section to correct for overestimation as a result of feature selection. Now that the independent validation has been added, correction for multiple testing is implicitly guaranteed. For any feature, the chance that it is statistically significant *and* in the same direction in two independent datasets, if there is no true association, is equal to $0.05 * 0.025 = 0.001$.

I. 177-182 I really find this problematic as the same results are published twice. I would have liked to see the results in one place.

Response: Please see response to first comment.

L 194 Do not use the OTU subscript notion. This is not common.

Response: We removed the subscript mode. OTUs are now referred to as e.g. *Enterococcus* (OTU1).

I. 278-280 These findings need to be validated as they are based on a limited number of CDI patients. I think this can be done for the same hospitals as included.

Response: Please see response to third comment.

I. 286-288. I doubt this would be possible, but if so then this test should have been used to validate the findings in the current work.

Response: The planned development of this novel method spans several years. For this reason, and because of the lack of leftover materials in the current study, we are, unfortunately, not able to perform such validation in the current work.

I. 327-328 I agree with the authors the low CDI incidence rate makes it difficult with prospective studies, including the risk of overfitting data based on few patients. I definitely think the study is big and valuable, but I am skeptical that the identified biomarkers are truly predictive in a general sense.

Response: Please see the results of the independent validation. We agree with the reviewer's concerns and had therefore carefully discussed the study limitations. Now that one of the models has been confirmed in an independent dataset, we think we can be more firm and have therefore rephrased our conclusion.

Reviewer #2 (Remarks to the Author):

This is a large, multi-center prospective cohort study (namely the ANTICIPATE study) that investigates the microbial signatures in patients with *Clostridioides difficile* infection (CDI). The overall goal of this project is to identify unique baseline biomarkers that predict the risk of CDI before receiving antibiotics.

The authors recruited 1,007 patients who underwent antibiotic therapy during hospitalization. Patients were followed for 90 days after infusion. Microbial (rectal swab and stool) samples were collected before/after antibiotics treatment and at the onset of a diarrhea episode. Of those, 1.14% and 1.89% of patients subsequently developed CDI within 28 days or 90 days, respectively. The patients who subsequently develop CDI had lower microbial diversity at the baseline. Perhaps not surprisingly, CDI patients carry toxigenic *C. difficile* before the onset of CDI. Among antibiotics, carbapenem prescription correlated with the incidence of CDI. Moreover, the ratio of *Enterococcus/Blautia* in the baseline microbiota was increased in patients who subsequently develop CDI. These parameters were not associated with the risk of non-CDI antibiotic-associated diarrhea (AAD), indicating that those are selective biomarkers for the prediction of CDI.

General comment:

Overall this is a well-performed clinical study that highlights potential biomarkers that can help predict the risk for CDI. In particular, microbial signatures associated with the risk of CDI are interesting. However, this study is correlative/descriptive, and hence the impact on the broader readership, particularly the basic science community, may be limited. Therefore, this manuscript is more suitable for a subspecialty clinical journal.

Response: Since the two manuscripts are mutually dependent and complementary, we have opted for a back-to-back publication.

Specific comments:

The authors nicely showed microbial biomarkers that may predict the risk of CDI before antibiotics treatment. The carriage of toxigenic *C. difficile* is not surprising, as this bacterium causes CDI. The increase in the ratio of *Enterococcus/Blautia* is interesting. However, the specificity and sensitivity of this microbial signature are not validated. It would be nice if the authors could validate their findings with other cohorts.

Response: We agree and thank the reviewer for this suggestion. We have now performed independent validation of the results using data from a previously published nested case-control study. Please see methods and results section for details. In short, the study includes 25 CDI patients and 25 randomly selected controls from a prospective cohort study. Microbiome analysis of fecal samples collected at baseline was performed in a way compatible with our study. The validation confirmed most of the candidate OTU models identified in our study. Some models were not confirmed, although the wide confidence interval of the relative risk estimates does not exclude an association either. We now selected the model based on two criteria, 1) being statistically significant in both datasets and 2) among these, the model that had the lowest AIC value. The OTU ratio model as well as the OTU abundance model performed similarly in the validation dataset as compared to our study.

REVIEWERS' COMMENTS

Reviewer #1 (Remarks to the Author):

I do not feel that the overlap issue between the head to head publications has not been properly solved. Still, in my head, the two papers should have been merged.

Reviewer #2 (Remarks to the Author):

The authors have addressed my previous concerns.

REVIEWERS' COMMENTS

Reviewer #1:

1. I do not feel that the overlap issue between the head to head publications has not been properly solved. Still, in my head, the two papers should have been merged.

Response: We thank the reviewer for reassessing the manuscript. Per decision of the editor we have kept the two manuscripts separated and have more clearly indicated the presence of overlapping data and analyses where it was deemed necessary to include results in both manuscripts.

Reviewer #2:

1. The authors have addressed my previous concerns.

Response: We thank the reviewer for reassessing the manuscript and for accepting our revisions.